# Ophthalmic Antimicrobial Prescribing in Australian Healthcare Facilities

**DOI:** 10.3390/antibiotics11050647

**Published:** 2022-05-12

**Authors:** Xin Fang, Noleen Bennett, Courtney Ierano, Rodney James, Karin Thursky

**Affiliations:** National Centre for Antimicrobial Stewardship, Melbourne, VIC 3000, Australia; noleen.bennett@mh.org.au (N.B.); courtney.ierano@unimelb.edu.au (C.I.); rodney.james@mh.org.au (R.J.); karin.thursky@mh.org.au (K.T.)

**Keywords:** antimicrobial resistance, antimicrobial stewardship, antimicrobial prescribing surveillance, quality, safety, ophthalmology, eye, surgical prophylaxis

## Abstract

The National Antimicrobial Prescribing Survey (NAPS) is a web-based, standardized tool, widely adopted in Australian healthcare facilities to assess the reasons for, the quantity of, and the quality of antimicrobial prescribing. It consists of multiple modules tailored towards the needs of a variety of healthcare facilities. Data regarding ophthalmological antimicrobial use from Hospital NAPS, Surgical NAPS, and Aged Care NAPS were analysed. In Hospital NAPS, the most common reasons for inappropriate prescribing were incorrect dose or frequency and incorrect duration. Prolonged duration was also common in Aged Care prescribing: about one quarter of all antimicrobials had been prescribed for greater than 6 months. All three modules found chloramphenicol to be the most prescribed antimicrobial with a high rate of inappropriate prescribing, usually for conjunctivitis.

## 1. Introduction

The National Antimicrobial Prescribing Survey (NAPS) is a web-based, standardised antimicrobial prescribing auditing programme [1]. The NAPS supports many of the objectives of the Australian National Antimicrobial Resistance Strategy [2], particularly with respect to implementing effective antimicrobial stewardship practices, and ensuring the judicious and appropriate use of antimicrobials. It also assists in the implementation of coordinated national surveillance of antimicrobial-use data, which contributes to the Antimicrobial Use and Resistance in Australia (AURA) Surveillance System reports [3].

The NAPS programme delivers insights into the quantity as well as the quality of antimicrobial prescribing at a facility, jurisdictional, and national level. Since the launch of the Hospital NAPS in 2013, the programme has diversified and grown into a programme that has identified target areas for improvement and supports the challenges of antimicrobial stewardship across Australian hospital and aged care settings, with the release of the Aged Care NAPS in 2015 and the Surgical NAPS in 2016.

Topical antimicrobial use in ophthalmology remains controversial, with limited evidence and indications for use as per the nationally endorsed prescribing guidelines [4]. From previous data analysis, it has been demonstrated that topical antimicrobial prescribing in ophthalmology often does not follow established guidelines, resulting in antimicrobials being prescribed widely for minor, self-limiting conditions and as post-surgical prophylaxis [5,6]. There is also limited literature that reports the extent and appropriateness of topical antimicrobial use in ophthalmology. Thus, highlighting a need to investigate such use. With between 4 and 8 years of data across the three NAPS datasets, this paper aims to identify common prescribing trends of topical ophthalmological antimicrobial use in various Australian healthcare settings.

## 2. Results

### 2.1. NAPS in Australia

Across the three NAPS modules, data for 261,784 antimicrobial prescriptions were analysed. Chloramphenicol was the most commonly prescribed topical antimicrobial for ophthalmology indications and surgical procedures across all three datasets (Hospital NAPS, 84.1% *n* = 1163; Aged Care NAPS, 92.6% *n* = 1107; and Surgical NAPS, 81.6 *n* = 1231%).

### 2.2. Hospital NAPS

The Hospital NAPS dataset (2013–2020) included 199,819 prescribed antimicrobials from 595 public and private hospitals, representing 61.6% of all Australian hospitals. Of all prescriptions, 1.2% (*n* = 2247) were for an ophthalmological indication. Of these, 61.5% (*n* = 1383) were administered via the topical ocular route. Chloramphenicol was the most prescribed antimicrobial, accounting for 84.1% of all topical ocular prescriptions (1163/1383).

When assessing appropriateness, 22.2% (*n* = 499/2247) of the ophthalmological prescriptions were assessed as inappropriate by the auditors. Among those, 4% (*n* = 89/2247) were considered inappropriate as they did not have an indication that warranted antimicrobial therapy. Auditors were unable to assess whether the indication required antimicrobial therapy in 23% (*n* = 516/2247) of the prescriptions. For the remaining 73.1% (*n* = 1642/2247) where antimicrobial therapy was indicated, the reasons for inappropriate prescribing are shown in Figure 1.

Incorrect dose or frequency was the most common reason for prescriptions to be assessed as inappropriate (*n =* 254), out of which 63% (*n =* 160) were prescribed for bacterial conjunctivitis. The second most common reason for inappropriate prescribing was assessed to be incorrect duration. The most common indication assessed as having incorrected duration was once again bacterial conjunctivitis (71.8%, *n =* 130).

Chloramphenicol prescriptions had a high rate of inappropriateness at 24.5%. Other antimicrobials with high rates of inappropriate use include doxycycline and ceftriaxone as shown in Figure 2.

### 2.3. Aged Care NAPS

The Aged Care NAPS dataset (2016–2020) included 20,484 prescribed antimicrobials from 1055 public, private, and not-for-profit aged care homes, representing 39.1% of all Australian aged care homes. Of these antimicrobial prescriptions, 6.3% (*n* = 1283 for 1171 residents) were for ophthalmic indications. The topical route accounted for 93.2% of all ophthalmologically indicated prescriptions (*n* = 1196/1283). Approximately one quarter (24.7%, *n* = 316) of these ophthalmic antimicrobials had been prescribed with a duration of greater than six months. The indication and review and stop dates were respectively documented for 75.1% (*n* = 964) and 24.6% (*n* = 315) prescriptions. Chloramphenicol accounted for the majority of the topical antimicrobial prescriptions (92.6%, *n* = 1107) and of these prescriptions, the most common indication was conjunctivitis (70.0%, *n* = 783).

Over the official timeframe, 5574 residents presented with one or more infections on the survey day. Of these residents, 6.5% (*n* = 363) were reported as having suspected conjunctivitis; 303 (83.5%) of the cases were confirmed as meeting the McGeer et al. surveillance definition [7].

### 2.4. Surgical NAPS

The Surgical NAPS dataset (2016–2020) included 46,766 prescribed antimicrobials (as procedural doses or post-procedural prescriptions for surgical prophylaxis), from 272 public and private hospitals, representing 31.2% of all Australian hospitals performing surgical procedures. Of this, 7.8% (*n* = 3630) were prescribed for ophthalmological procedures. Topical antimicrobials accounted for almost half (41.6%) of all antimicrobial use in ophthalmological surgery (*n* = 1510). The majority of the antimicrobials were prescribed post-procedurally (*n* = 1047, 69.3%). Chloramphenicol was the most prescribed topical antimicrobial both procedurally (91.1%, *n* = 422) and post-procedurally (77.3%, *n* = 809).

Overall, 42.4% of all topical ocular antimicrobial prescriptions were deemed non-compliant with guidelines. Non-compliance was higher when comparing procedural and post-procedural prescribing (48.8% and 39.6% respectively). Compliance with the nationally endorsed prescribing guidelines [4] was 36.3%, however, this differed between procedural and post-procedural prescribing (21.4% vs. 42.9% respectively). Local guideline compliance accounted for 10.3% of prescriptions.

When assessing appropriateness, 48.6% (*n* = 734/1510) were deemed appropriate and 39.8% (*n* = 601/1510) were deemed inappropriate. The most common reason for inappropriate procedural doses was that an antimicrobial was not indicated for use 65.9% (*n* = 149 of 226 reasons). Prolonged duration of prophylaxis accounted for 73% of all reasons for inappropriate post-procedural prescriptions (*n* = 287/395).

Of all post-procedural prescriptions (*n* = 1047), 89.5% (*n* = 937) had a documented duration ranging from zero to 37 days with a median of 10 days as shown in Table 1. Appropriateness is reduced after a duration of 7 days and this is likely to reflect the national guidelines recommendation of permitting up to 7 days of topical ocular chloramphenicol post ophthalmological procedures [4].

## 3. Discussion

There is limited literature reporting on the extent and appropriateness of antimicrobial prescribing in ophthalmology. The NAPS programme aims to address this recognized gap to support targeted AMS initiatives. It is important that assessments of antimicrobial prescribing quality are conducted globally. This will provide valuable insight at the local level whilst also facilitating meaningful benchmarking at the facility, regional, and national levels. Our analysis of the Hospital and Aged Care NAPS data demonstrated that conjunctivitis was the most common ophthalmic infection indicated for antimicrobial use within the hospital and aged care sectors. Chloramphenicol was the most common topical ophthalmological antimicrobial prescribed across all three NAPS datasets, with varying rates of guideline compliance and appropriateness. Table 2 summarises the clinical indications in which chloramphenicol is and is not recommended as per the nationally endorsed guidelines (4).

Conjunctivitis is a common medical condition that is usually managed in primary care. Conjunctivitis may be allergic, viral, or bacterial in etiology. Allergic conjunctivitis affects 15–40% of the population and the symptoms are often seasonal [8]. Viral conjunctivitis is the most common type of infectious conjunctivitis, accounting for 20% to 70% of infections with the majority (65–90%) being caused by adenovirus [9]. Treatment for viral conjunctivitis is usually supportive. It does not require antimicrobial therapy and may include cold compresses, artificial tears, and topical antihistamines [9].

Bacterial conjunctivitis is the second most common cause of infectious conjunctivitis and is of higher prevalence in the paediatric population [10]. Acute infective conjunctivitis is rarely a cause of vision loss and is generally self-limiting. Although systematic reviews have observed that topical antibiotics may confer a modest benefit in hastening early symptom resolution in bacterial conjunctivitis, most cases resolve spontaneously within 7 days [11,12]. However, topical antimicrobial therapy may be reasonable in patients with mucopurulent symptoms and treatment should always be initiated in neonates [13]. The national guidelines in Australia have incorporated the above evidence into their recommendations (4) and recommend up to 7 days of chloramphenicol in patients who present with marked symptoms, such as purulent discharge. The NAPS survey does not require auditors to assess the accuracy of the diagnosis and viral conjunctivitis and allergic conjunctivitis could have been included as bacterial conjunctivitis. Furthermore, chloramphenicol prescriptions assessed as appropriate might have been prescribed for bacterial conjunctivitis with mild symptoms which could have resolved spontaneously without antimicrobial treatment.

In Hospital NAPS, chloramphenicol was the most commonly prescribed antimicrobial. Therefore, targeting its use will have the most significant impact on the quality of antimicrobial prescribing. The most common reasons for inappropriate prescribing were inappropriate dose or frequency and incorrect duration. Bacterial conjunctivitis was the main indication contributing to both of those categories of inappropriate prescribing. This suggests that antimicrobial stewardship (AMS) activities targeting dosing and duration of chloramphenicol in treating bacterial conjunctivitis will likely have a considerable impact on improving ophthalmic antimicrobial prescribing in hospitals.

In Aged Care NAPS, auditors were not required to assess the appropriateness of prescribing. However, 783 chloramphenicol prescriptions were prescribed to treat conjunctivitis, but only 363 residents had suspected conjunctivitis (303 of which met the McGeer et al. definition) on the day of the survey. A possible explanation for this is that some of the residents’ symptoms may have subsided on the audit day. However, by large this highlights the high probability of overprescribing chloramphenicol in suspected and confirmed cases of conjunctivitis. In addition, it was very concerning that 24.7% of ophthalmic antimicrobials were prescribed with a duration of greater than six months, highlighting a large-scale lack of antimicrobial review among Australian aged care facilities. We advocate for the adoption of general antimicrobial stewardship principles [14] in which, all antimicrobials are prescribed for the shortest possible duration of therapy, consistent with the condition being treated and the patient’s clinical response. As an AMS initiative, the review or stop dates should be recorded at the time of initial antimicrobial prescribing in aged care facilities.

Prior to 1 May 2010, Chloramphenicol was a prescription-only (schedule 4) medication in Australia [15]. It is now available as an over-the-counter (schedule 3) medication. This means that patients in the community can purchase it from a pharmacy without a prescription. Data suggest that there has been a significant increase in chloramphenicol use post schedule change, which is likely caused by its overuse in presumed bacterial conjunctivitis. Overprescribing of chloramphenicol in the community setting is further supported by a Registrar Encounters in Clinical Training (ReCEnT) study [16] where it was found that antimicrobials (mainly topical) were prescribed at what was considered an excessive rate of 74% of all diagnoses (*n =* 1170) of infective conjunctivitis. As with our NAPS findings, chloramphenicol was the most commonly prescribed topical antimicrobial (95.8%).

Our analysis of the Surgical NAPS datasets has also highlighted the problematic use of topical ophthalmological antimicrobials. The use of topical antimicrobial prophylaxis is not routinely recommended as per national guidelines for surgical prophylaxis (see Table 2) [4,14]. An exception to this is in ophthalmological procedures, where short-term post-operative topical antimicrobial prophylaxis may be appropriate for a maximum of 7 days if considered necessary. Our findings are reflective of this caveat in the guidelines, as appropriateness was notably higher (80.1%) in post-procedural prescriptions with a duration of less than 7 days compared to those with a duration of greater than 7 days (48.2%). Procedural appropriateness remained considerably high (48.2%); this may also be resultant of this caveat that permits 7 days of topical antimicrobial use inclusive of the procedural dose. However, we questioned whether the indication truly warranted topical antimicrobial use in the first place. Greater education and support for auditors is required to ensure the accuracy of appropriateness assessments and, similarly, of prescribers to ensure accurate documentation of antimicrobial indications.

Cataract surgery is the most frequently performed elective procedure worldwide. With no concomitant eye disease, there is usually an excellent prognosis following surgery. Endophthalmitis is a rare but serious, sight-threatening complication that can occur following intraocular surgeries, mainly cataract surgery [17]. Although there is demonstrated evidence that antibiotics injected intracamerally (into the anterior chamber of the eye) at the end of cataract surgery are effective in reducing the incidence of endophthalmitis, there is a lack of convincing evidence to support the administration of post-operative topical antimicrobials, in particular chloramphenicol [17,18,19]. Furthermore, some recent studies have suggested that post-operative topical antibiotic drops may, in fact, increase the risk of endophthalmitis, due to the incorrect technique of patient-administered eyedrops. Such complications may include corneal or conjunctival abrasions and the potential for wound gape and efflux of fluid into the eye following pressure on the eye [20]. Despite such evidence, many surgeons continue to prescribe antibiotics after intravitreal or intracameral injection. This is further illustrated by the Surgical NAPS data, in which, 64% of all post-procedural prescriptions had a duration greater than 8 days (Table 1). Other applications of topical antimicrobial prophylaxis (i.e., non-ocular use) appear to remain a common practice among surgeons despite limited evidence. The rationale for ongoing antimicrobial use in ophthalmic procedures is not clear, current national guidelines recommend “cefazolin 1 mg/0.1 mL intracamerally, as a single dose at the end of cataract surgery” [4]. However, the inclusion of an exception-like statement that suggests “short term post-operative topical antimicrobial prophylaxis may be appropriate for a maximum of 7 days, if considered necessary” [4] may be an influential driver for prolonged antimicrobial duration. We advocate for qualitative research to explore the decision-making processes behind ophthalmic surgeons’ antimicrobial prescribing and whether the current guidelines are a relevant and influential resource for such prescribing behaviours.

Identifying topical antimicrobial prophylaxis use in surgical procedures in ophthalmology and other specialties represents a niche target for antimicrobial stewardship programmes. For example, the Australian Commission on Safety and Quality in Health Care has published case studies for their Cataract Clinical Care Standard [5]. One of these case studies highlighted an Australian hospital’s Surgical NAPS audits revealing high rates of inappropriate use of postoperative topical antibiotics in ophthalmology. The findings were fed back to the Head of the Infectious Diseases unit who alerted the Head of Ophthalmology. Subsequently, the Ophthalmology department adopted a change in practice and ceased the routine prescription of topical chloramphenicol. This led to a notable reduction in chloramphenicol use at their local service. The above case study provides a novel application of the Surgical NAPS data to facilitate a change in surgical antimicrobial prescribing behaviour. We advocate that this approach may be translatable across multiple hospital settings and support further optimization of antimicrobial use in ophthalmological surgery nationwide. It could be particularly effective when AMS initiatives are agreed upon and implemented using a “top down” approach with strong support from hospital executives and departmental heads. The progressive uptake of electronic medical record systems provides further opportunities to support AMS. For example, to address prolonged antimicrobial prescribing, the electronic medical record system may introduce an automated alert for specific duration lengths or mandate the inclusion of a review or cessation date. Such processes would notify the prescriber or AMS team of patients with prolonged antimicrobial prescriptions and prompt further clinical reviews of the antimicrobial therapy.

## 4. Conclusions

The NAPS data demonstrated that conjunctivitis was the most common ophthalmic infection indicated for antimicrobial use within the hospital and aged care sectors. Chloramphenicol was the most common topical ophthalmological antimicrobial prescribed across all three NAPS datasets. AMS activities should focus on reducing inappropriate dosing, frequency, and duration when prescribing chloramphenicol, as well as avoiding unnecessary and prolonged chloramphenicol prescribing in surgical prophylaxis.

## 5. Materials and Methods

De-identified data for analysis were extracted from the NAPS database between 2013–2020 for Hospital NAPS, 2016–2020 for Surgical NAPS, and 2015–2020 for Aged Care NAPS. Data were included for the indications listed in Table A1. Every Australian state and territory, funding type, peer group classification, and remoteness area are represented in their respective NAPS modules. Auditors encompassed the spectrum of health care professionals including doctors, pharmacists, nurses, infection prevention and control practitioners, and quality managers.

The Hospital NAPS is a point prevalence survey available year-round, and hospitals are encouraged to complete the survey at least once every calendar year. Data collected includes admitted specialty, prescribed antimicrobials, dose, route, frequency, and indication for antimicrobial use. Relevant clinical notes, including any comorbidities, renal function, history of antimicrobial allergy, and relevant microbiology results are also collected. The auditing team determines the guideline compliance and appropriateness of prescribing based on the nationally endorsed prescribing guidelines (Therapeutic Guidelines [4]) and any locally endorsed guidelines.

The Surgical NAPS is a period prevalence survey, available year-round, with flexible methodologies and can be performed retrospectively or prospectively, allowing auditors to target surgical craft groups or procedures of interest. The auditing team is required to assess the guideline compliance and appropriateness for both procedural and post-procedural surgical antimicrobial prophylaxis. Data collected includes procedures performed, times of incision and end of surgery, prescribed antimicrobials (both procedurally and post-procedurally) dose, route, frequency, and administration time. Relevant clinical notes including removal or insertion of prosthetic material, risk factors, relevant microbiology results, allergy status, and any existing antimicrobial therapy are also collected.

The Aged Care NAPS is typically open for a period of 3–6 months annually. Participation is voluntary with the exception of Victoria public aged care homes, for whom participation is mandated by the State Department of Health. Data on residents prescribed an antimicrobial on the survey day are collected. This includes antimicrobial selection, start date if known and less than 6 months (otherwise greater than 6 months or unknown), frequency including pro re nata (PRN), route of administration, therapy type (prophylaxis or therapeutic), and indication. The indication for antimicrobial prescribing is reported according to a standardized list.

Additionally for the Aged Care NAPS, data on residents with signs and symptoms of infections on the survey day or in the two days prior are collected. The signs and symptoms are categorized according to six body systems: urinary tract, respiratory tract, skin or soft tissue, oral, eye, and other. Algorithms were used during the analysis stage to determine if the signs and/or symptoms meet the McGeer et al. surveillance definitions [7]

Although participation is generally voluntary, the NAPS is well adopted with over 60% of Australian hospitals and up to 30% of aged care facilities participating annually. This is due to the ability of NAPS to support and enhance local antimicrobial stewardship initiatives while assisting facilities in meeting their requirements for national accreditation standards for both the hospital and aged care sectors. The NAPS also has the added benefit of real-time, automated reporting with benchmarking, which allows for immediate feedback to prescribers, to assist with local antimicrobial stewardship initiatives. In contrast to other published antimicrobial auditing tools, the Hospital and Surgical NAPS are unique in allowing for the assessment of the quality of a prescription. In addition to whether prescriptions are concordant with prescribing guidelines, the surveys also consider any documented, clinically justifiable reasons to vary from the recommended guidelines designated as ‘appropriate’ or ‘inappropriate’ prescribing.

All modules are accompanied by education materials including data collection forms, user guides, and eLearning modules to ensure data standardization and validity. Appropriateness assessment guides are available for Hospital and Surgical NAPS. The NAPS support team is available to provide additional clinical advice and assessments of appropriateness for facilities without infectious diseases specialists or antimicrobial stewardship leaders or experts.

## Figures and Tables

**Figure 1 antibiotics-11-00647-f001:**
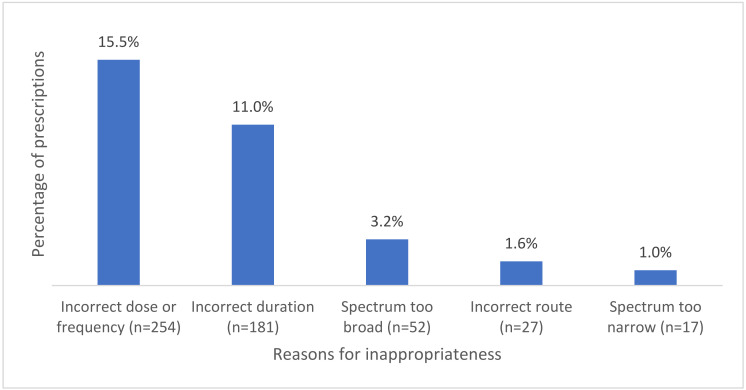
Common reasons for inappropriateness where antimicrobials were indicated.

**Figure 2 antibiotics-11-00647-f002:**
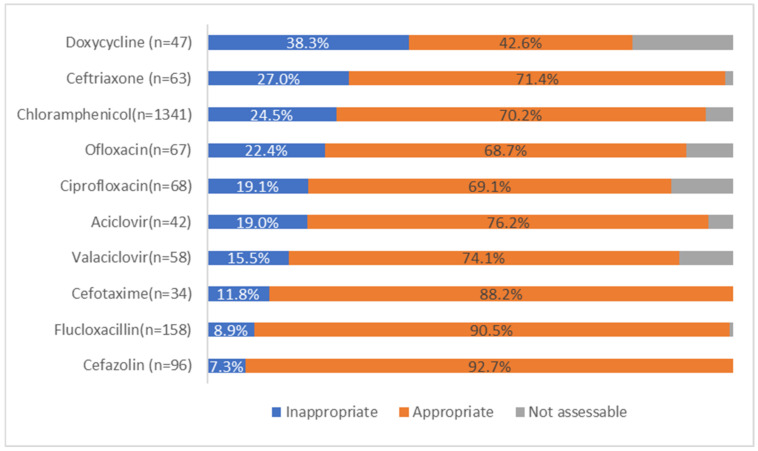
Appropriateness for the 10 most prescribed ophthalmological antimicrobials.

**Table 1 antibiotics-11-00647-t001:** Duration and appropriateness of post-procedural prescriptions.

Post Procedural Duration	Appropriateness *n* (%)	Total (*n*)
Days 0–7	282 (80.1)	352
Days 8–37	282 (48.2)	585
Total	564 (60.2)	937

**Table 2 antibiotics-11-00647-t002:** Summary of indications for chloramphenicol use as per the nationally endorsed guidelines, the Therapeutic Guidelines [4].

Use Indicated	Use Not Indicated
Bacterial conjunctivitis	Viral conjunctivitis
Prophylaxis for corneal abrasions	Venous leg ulcers
Prophylaxis for removal of corneal foreign bodies	Exposed tissue/open wounds/ulcers
Post-procedural prophylaxis for ophthalmological surgery	Procedural surgical prophylaxis
Anterior blepharitis	

## Data Availability

All de-identified data related to this study may be available and accessible on request from the corresponding author.

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
