# Peer review of "Ophthalmic Antimicrobial Prescribing in Australian Healthcare Facilities"

_antibiotics, 2022, doi:10.3390/antibiotics11050647_

Round 1
Reviewer 1 Report
The Authors investigated ophthalmic antimicrobial prescribing in Australian healthcare facilities. They found a high rate of inappropriate prescribing, usually for conjunctivitis.
The manuscript highlights the problem of inappropriate antibiotic prescribing in Ophthalmology, which contributes to the phenomenon of increasing antibiotic-resistance.
The Authors' results are important and should reach the largest possible readership. Appropriate dosing, frequency, and duration of topical antibiotic therapy in Ophthalmology are of critical importance to reduce the risk of selection of multi-drug-resistant bacterial strains
- What is the main question addressed by the research?
To investigate antimicrobial prescribing in Australian healthcare facilities.
2. Do you consider the topic original or relevant in the field? Does it address a specific gap in the field?
The topic is relevant in Medicine in general, and in Ophthalmology in particular. Topical antimicrobial use in Ophthalmology remains controversial, with limited evidence and indications. Inappropriate use of topical antibiotics contributes to the selection of multi-resistant bacterial strains.
3. What does it add to the subject area compared with other published material?
There is limited literature reporting the extent and appropriateness of topical antimicrobial use in Ophthalmology. The results of this study highlights that the most common reasons for inappropriate prescribing were incorrect dose/frequency and incorrect duration. Prolonged duration was also common in Aged Care prescribing (about 25% of all antimicrobials had been prescribed for >6 months). Chloramphenicol was the most prescribed antimicrobial with a high rate of inappropriate prescribing, usually for conjunctivitis.
4. What specific improvements should the authors consider regarding the methodology? What further controls should be considered?
The Methodology is appropriate.
Are the conclusions consistent with the evidence and arguments presented and do they address the main question posed?
Yes. However, the need for similar investigations in other areas of the World should be emphasized.
6. Are the references appropriate?
Yes
7. Please include any additional comments on the tables and figures.
Tables and Figures are appropriate
Author Response
Thank you very much for reviewing our submission and your valuable insights.
We agree that there is a need for similar investigations to be conducted across regions in the world. We have added a paragraph in discussion to highlight this need.
Reviewer 2 Report
This was a clearly written paper which provides much needed analysis of information collected around antibiotic use in Australia. The analysis included hospitals and aged care setting which is crucial to understadning the wider picture of prescribing and ultimately AMR. It would be beneficial to add in the conclusions one or two specific AMS strategies for opthalmologists to adopt.
Author Response
Thank you very much for your review and insights.
We agree adding a couple of specific AMS strategies will be beneficial for the readers. We have made two specific suggestions:
- Implement AMS initiatives using a "top down" approach, starting with departmental heads (e.g. ID and ophthalmology) agreeing on an initiative. This will ensure junior doctors are not put in a position where they feel they lack the authority to challenge the norm of prescribing which might be inappropriate.
-
Automated alters or cessation of an order. In the electronic medication management system, the prescriber will be notified of patients with prolonged antimicrobial prescriptions and therefore prompted to cease the order while conducting a patient review.
Reviewer 3 Report
The paper describes the ophthalmic antimicrobial prescribing in hospital, surgical and aged care settings in Australia. I only have a minor comment:
p-6 line 211-213 "Despite such evidence, many surgeons...."Please clarify, if this is true because of lack of post-surgery prescribing guidelines or there is any other reason.
Author Response
Thank you very much for your review and insight.
We have added the following in discussion to address your question:
The rationale for ongoing antimicrobial use in ophthalmic procedures is not clear, current national guidelines recommend 'cefazolin 1 mg/0.1 mL intracamerally, as a single dose at the end of cataract surgery' [4]. However, the inclusion of an exception-like statement that suggests 'short term post-operative topical antimicrobial prophylaxis may be appropriate for a maximum of 7 days, if considered necessary' may be an influential driver for prolonged antimicrobial duration. We advocate for qualitative research to explore the decision-making processes behind ophthalmic surgeon's antimicrobial prescribing and whether the current guidelines are a relevant and influential resource to such prescribing behaviors.